# An Anti-Invasive Role for Mdmx through the RhoA GTPase under the Control of the NEDD8 Pathway

**DOI:** 10.3390/cells13191625

**Published:** 2024-09-28

**Authors:** Lara J. Bou Malhab, Susanne Schmidt, Christine Fagotto-Kaufmann, Emmanuelle Pion, Gilles Gadea, Pierre Roux, Francois Fagotto, Anne Debant, Dimitris P. Xirodimas

**Affiliations:** 1CRBM, Cell Biology Research Centre of Montpellier, Université de Montpellier, CNRS, 34293 Montpellier, France; susanne.schmidt@crbm.cnrs.fr (S.S.); christine.fagotto-kaufmann@crbm.cnrs.fr (C.F.-K.); emmapion@googlemail.com (E.P.); gilles.gadea@inserm.fr (G.G.); pierre.roux@crbm.cnrs.fr (P.R.); francois.fagotto@crbm.cnrs.fr (F.F.); 2Research Institute of Medical and Health Sciences, University of Sharjah, Sharjah 27272, United Arab Emirates

**Keywords:** cell migration, cell invasion, Mdmx, NEDD8, RhoA

## Abstract

Mdmx (Mdm4) is established as an oncogene mainly through repression of the p53 tumour suppressor. On the other hand, anti-oncogenic functions for Mdmx have also been proposed, but the underlying regulatory pathways remain unknown. Investigations into the effect of inhibitors for the NEDD8 pathway in p53 activation, human cell morphology, and in cell motility during gastrulation in Xenopus embryos revealed an anti-invasive function of Mdmx. Through stabilisation and activation of the RhoA GTPase, Mdmx is required for the anti-invasive effects of NEDDylation inhibitors. Mechanistically, through its Zn finger domain, Mdmx preferentially interacts with the inactive GDP-form of RhoA. This protects RhoA from degradation and allows for RhoA targeting to the plasma membrane for its subsequent activation. The effect is transient, as prolonged NEDDylation inhibition targets Mdmx for degradation, which subsequently leads to RhoA destabilisation. Surprisingly, Mdmx degradation requires non-NEDDylated (inactive) Culin4A and the Mdm2 E3-ligase. This study reveals that Mdmx can control cell invasion through RhoA stabilisation/activation, which is potentially linked to the reported anti-oncogenic functions of Mdmx. As inhibitors of the NEDD8 pathway are in clinical trials, the status of Mdmx may be a critical determinant for the anti-tumour effects of these inhibitors.

## 1. Introduction

The Mdm2 homologue Mdmx (also known as Mdm4) is a critical negative regulator of the p53 tumour suppressor. Mdmx directly interacts with p53 and negatively controls p53 function in a mutually exclusive manner of its homologue Mdm2 [1,2,3]. In addition, Mdmx potentiates Mdm2 ubiquitin E3-ligase function through the formation of hetero-oligomers via their RING domains [4,5]. In parallel, Mdmx is an Mdm2 substrate for ubiquitin-mediated proteasomal degradation [6,7]. Increased expression in Mdmx due to gene amplification is observed in multiple types of cancers. However, in several cases, increased levels of Mdmx are due to post-transcriptional events. In over 60% of cutaneous melanoma, wild-type p53 is inactivated due to high protein levels of Mdmx in the absence of deregulation at the mRNA level [8]. While the above studies emphasised the important role of Mdmx in repressing p53 function as part of its oncogenic properties, an increasing body of evidence suggests that Mdmx has p53-independent functions [9]. Interestingly, amongst these studies, it was shown that Mdmx represses the transformation and tumour progression of p53 null cells in vitro and in vivo, indicating a p53-independent tumour suppression role for Mdmx [10]. However, the molecular basis for these new and intriguing functions of Mdmx remains unknown.

The covalent attachment of ubiquitin and ubiquitin-like molecules (Ubls) to target proteins plays an important role in a variety of cellular biological processes. Among the family of Ubls, NEDD8 is closest to ubiquitin, to which it is 60% identical and 80% homologous [11,12]. Upon its synthesis, NEDD8 is processed by NEDP1 (DEN1 or SENP8) and, similarly to ubiquitination, is conjugated to substrate proteins through a multistep enzymatic cascade, involving E1, E2, and E3 enzymes [13]. While many potential targets for NEDDylation have been identified, the best characterised substrate is the Cullin family of proteins [14,15,16,17]. Cullins are molecular scaffolds of the largest class of RING-type E3 ligases (Cullin-Ring-Ligases, CRLs). CRLs promote the ubiquitination of about 20% of cellular proteins targeted for proteasomal degradation [18,19]. NEDDylation of Cullins increases the activity of CRLs, thus promoting the ubiquitination and proteasomal degradation of the target substrates [18]. The NEDD8 pathway regulates many cancer-related processes, including cell cycle progression, DNA replication, and repair, and is upregulated in tumours [11,20]. The NEDD8 inhibitor MLN4924 (Pevonedistat), which selectively targets the NEDD8 E1-activating enzyme, has shown a strong anti-cancer response in pre-clinical studies and is now in Phase III clinical trials [11,21]. Suppression of cell migration and invasion are key effects related to the anti-cancer properties of NEDD8 inhibitors or of natural medicinal compounds such as the alkaloid Sanguinarine [22,23,24,25,26,27,28]. The small GTPase RhoA was found to be a critical regulator of the above-described effects induced upon NEDDylation inhibition [24]. RhoA belongs to the family of Rho GTPases that are involved in major functions, such as cell polarity, adhesion, motility, and differentiation, by their capacity to control actin cytoskeleton remodelling [29,30]. It is, therefore, not surprising that the Rho GTPase signalling also mediates multiple aspects of tumour progression, including cell transformation, invasion, and metastasis [30,31]. RhoA is specifically renowned for its ability to promote the assembly of actin stress fibers and focal adhesions and to drive actomyosin-based membrane blebbing and contractility [31]. Importantly, a fine regulation of RhoA-dependent actomyosin contractility is required to control cell migration; contractility is needed for effective anchoring, traction and polarisation, but high contractility rigidifies the cell cortex and antagonises motility [32]. Similarly to all of the Rho GTPases, the activity of RhoA is finely tuned by different regulators [33,34]. Guanine nucleotide exchange factors (GEFs) activate GTPases by promoting the exchange of GDP to GTP, whereas GTPase-activating proteins (GAPs) inactivate GTPases by enhancing their intrinsic GTPase activity [35,36,37,38]. Moreover, guanine nucleotide dissociation inhibitors (GDIs) stabilise cytosolic GTPases by forming soluble complexes with GDP-bound Rho GTPases and participate to the membrane targeting of the Rho GTPases [35,36,37,38]. In addition to the tight regulation of its activity, RhoA is also controlled at the protein stability level by the ubiquitin–proteasome system. RhoA is a substrate of the E3 ligases CRL1, CRL3, and Smurf1, which target RhoA for ubiquitin-mediated proteasomal degradation. Inhibition of the NEDD8 pathway blocks RhoA degradation through the inactivation of CRLs and, potentially, Smurf1 [35,36,37,38]. The resulting stabilisation of RhoA was proposed as the key mechanism responsible for the anti-invasive effect of NEDD8 inhibition by MLN4924 [24].

In this study, by investigating the effect of inhibitors of the NEDD8 pathway in p53 activation, human cell morphology, and cell motility during gastrulation in Xenopus embryos, we found Mdmx to be a critical component of the NEDD8-RhoA module and cell invasion process. Mdmx protects RhoA from degradation and is required for the anti-invasive effect of NEDD8 inhibition in metastatic cells. Biochemical analysis shows that Mdmx directly binds preferentially to the GDP form of RhoA through its Zn finger domain. The above-described roles of Mdmx are independent of wild-type p53 function. Prolonged inhibition of NEDDylation causes a late degradation of Mdmx. This step requires the presence of non-NEDDylated Cullin4A and the E3-ligase Mdm2, and directly impacts the RhoA protein level, resulting in RhoA destabilisation. Our studies reveal a previously uncharacterised role of Mdmx in cell invasion through the RhoA GTPase, providing mechanistic insights into the emerging tumour suppressor functions of Mdmx.

## 2. Materials and Methods

### 2.1. Chemicals

All common chemicals were purchased by VWR and Sigma Aldrich. MLN4924 was purchased from Active Biochem (A-1139), Y27632 (TOCRIS), MG132 from Viva Bioscience, trypsin from Promega, Madison, WI, USA.

### 2.2. Antibodies

Mouse anti-HA (1:2000, Princeton, NJ, USA), mouse anti-Mdmx (1:1000, Millipore, Burlington, MA, USA), 4B2 mouse anti-Mdm2 (1:1000, in house), rabbit anti-Cullin4A (1:1000, Abcam, Cambridge, UK), rabbit monoclonal anti-NEDD8 (1:2000, Abcam), mouse anti-RhoA (1:250, Santa Cruz, CA, USA), DO-1 mouse anti-p53 (1:1000, in house), rabbit anti-p21 (1:250, Santa Cruz), mouse anti-α-Tubulin (1:2000, Cell Signaling, Danvers, MA, USA), mouse anti-MLC2 (1:2000, Sigma-Aldrich, Burlington, MA, USA), rabbit phospho-MLC2 (1:1000, Cell Signaling), mouse anti-flotillin (1:1000, 610820, BD Transduction laboratories, Bergen County, NJ, USA), rabbit anti-Ubiquitin (1:2000, DAKO, Santa Clara, CA, USA), mouse anti-GAPDH (1:1000, Santa Cruz), mouse anti-β-actin (1:2000, Calbiochem, San Diego, CA, USA), rabbit anti-Ube2M (1:2000, Abgent, San Diego, CA, USA), rabbit monoclonal anti-Ube2F (1:1000, Abcam), mouse anti-Flag M2 (1:2000, Sigma-Aldrich), rabbit monoclonal anti-RhoC (1:1000, Cell Signaling), mouse anti-Tubulin (Cell Signaling), rabbit anti-Rac1 (Cell Signaling), rabbit anti-H2A (1:2000, Abcam), and rabbit anti-GDI1 (1:1000, Santa Cruz).

### 2.3. Cell Lines and Growth Conditions

A375 human melanomas, MDA-MB-231 metastatic breast cancer cells were cultured in Dulbecco’s Modified Eagle’s Medium (DMEM, Gibco, New York, NY, USA) including 50 µg/mL Streptomycin, 50 U/mL Penicillin, and 10% FCS (foetal calf serum) at 37 °C, 5% CO_2_. The 90% confluent cells were washed once with 37 °C PBS (Phosphate-Buffered Saline, Gibco), treated with trypsin-EDTA, and replated to the appropriate confluency. Cells were routinely monitored for mycoplasma contamination and replaced after they had reached passage 25.

### 2.4. Transfections

Cells were transiently transfected with plasmid DNA using the Fugene HD Transfection Reagent (Roche, Basel, Switzerland) according to the manufacturer’s instructions. Cells were seeded on 6-well plates or 10 cm dishes and transfected when they reached 70–80% confluency. Lipofectamine RNAiMAX (Invitrogen, Waltham, MA, USA) was used for siRNA transfections with 5 nM of siRNAs. siRNAs, including non-targeting siRNA, were purchased from Dharmacon as ON-TARGETplus SMARTpools (4 individual siRNAs for each gene).

### 2.5. Cell Cycle Analysis

Fluorescence-Activated Cell Sorting (FACS) was used to analyse the cell cycle distribution. Cells seeded in 6-well plates were harvested using trypsinisation and were centrifuged and resuspended in 1 mL of PBS. Fixation was performed in 70% ethanol and cells were pelleted before they were resuspended in 1 mL PBS containing RNAse A (10 µg/mL, Invitrogen) propidium iodide (20 µg/mL, Sigma-Aldrich, P4170), incubated at room temperature for 30 min. Samples were analysed using a Becton Dickinson FACScan with the CELLQuest software, version 5.1.

### 2.6. Immunofluorescence—Stress Fibers, RhoA Staining

Cells plated on coverslips at ~25% confluence were transfected the next day with different siRNAs and treated as indicated. Post-treatment, cells were washed 3 times with warm PBS and fixed with 4% paraformaldehyde for 5 min at room temperature. Cells were washed with ice-cold PBS (3 times) and permeabilized for 10 min with PBS containing 0.1% triton X-100%. Cells were washed 3 × 10 min with PBS, and then were treated with blocking buffer (PBS containing 1% goat serum and 0.05% tween) for 30 min at room temperature. Cells were incubated with the corresponding antibody diluted in 0.05% tween, for 1 h in Rhodamin Phalloidin used at 100 nM diluted in blocking buffer or overnight in mouse monoclonal anti-RhoA antibody (1:250 Santa Cruz Biotechnology, Santa Cruz, CA, USA). Cells were washed 3 × 10 min with PBS followed by incubation for 1 h with corresponding fluorescein (FITC)-conjugated secondary antibody (life technology) diluted (1:500) in 0.05% tween +1% Goat serum in PBS for 1 h in a dark chamber at room temperature (RhoA). Samples were washed 3 × 10 min with 0.05% tween in PBS and stained with DAPI (Sigma-Aldrich) used to mark nuclei compartment (1:20,000) for 20 s. Cells were washed 3 times with PBS, mounted with Vectashield Mounting Medium (H-1000, Vector Laboratories, CA, USA), and sealed. Images were visualised and captured using Zeiss Axioimager Z2 Microscope (Oberkochen, Germany). Images were analysed using ImageJ-win64 software. At least 100 cells per condition were used for quantification in three independent experiments.

### 2.7. Transwell Migration (Cell Invasion) Assay

BSA coating: 100 µL BSA (Sigma-Aldrich) solution was added per well and mixed with fluoro beads (1:1000) (molecular probe F13083) in a black 96-well ViewPlates (Perkin-Elmer, Waltham, MA, USA). Plates were centrifuged at 4000 rpm for 30 min at 4 °C. A concentration of 10^5^ cells/mL was prepared in serum-free liquid bovine collagen I at 2.3 mg/mL and 100 μL aliquots were added per well. Plates were then centrifuged at 1000 rpm for 5 min to force cells towards the bottom of the wells, and then were incubated in a 10% CO_2_ tissue culture incubator at 37 °C. Upon collagen polymerization, 1% or 5% FCS was added on top of the collagen with either MLN4924 or DMSO as control treatment. After 24 h incubation at 37 °C in 10% CO_2_, 20% formaldehyde (Sigma-Aldrich) solution containing 5 mg/mL Hoechst 33,258 was used for 2 h to fix and stain cells to achieve final formaldehyde concentration at 4%. Evaporation and light exposure was prevented by sealing the plates with black tape. Confocal z slices were collected from each well at 40 mm for counting invading cells and at the bottom of the wells (3 mm) for counting total cells with an INCELL3000 high-content microscope. Nuclear staining in each slice was quantified automatically using the INCELL3000 software (version 7.3.0) with the Object Intensity module for determination of the percentage of invasive cells. Samples were run in triplicate and mean values ± standard deviation are presented.

### 2.8. Monitoring Gastrulation in Xenopus Embryos

Xenopus laevis embryos were obtained from females injected with human chorionic gonadotropin and fertilised in vitro. The jelly coat was removed by treatment in 2.5% L-cysteine pH 8.0. For injections, embryos were put in 2% Ficoll PM400 in 1× MBSH (MBS-H (modified Barth solution containing: 88 mM NaCl, 1 mM KCl, 2.4 mM NaHCO_3_, 0.82 mM MgSO_4_, 0.33 mM Ca(NO_3_)_2_, 0.41 mM CaCl_2_, 10 mM HEPES, 10 μg/mL streptomycin sulphate and penicillin, pH 7.4 adjusted with NaOH). In total, 40 ng of morpholinos and 25 pg N19RhoA mRNA were injected in each of the 2 blastomeres of 2-cell stage embryos. Embryos were transferred and let develop in 1/10 MBSH. MLN4924 was diluted 1/500 from a stock 10 mM in DMSO to 20 μM in 1/10 MBSH. Inhibitor treatment was started at stage 9 (blastula) for a 6 h period in total. Control embryos were incubated in 0.2% DMSO. Images were taken at midgastrula stage (stage 11–12) using a Zeiss DV8 stereomicroscope equipped with a Camera HDMI TrueChrome Metrics colour digital camera (Tucsen, Fuzhou, Fujian, China). The blastopore area was measured using the ImageJ software (version 1.50i) and normalised to the total cross-section equatorial area of the embryo.

mRNA coding for the negative form T19N of Xenopus RhoA was synthesised from the corresponding pBSK vector using SP6 polymerase, following the manufacturer’s instructions (mMessage mMachine kit, Ambion, Austin, TX, USA). Xenopus Mdmx morpholino (ATGCAAAGCAGTTGATGTAGACATG) and control morpholino (CCTCTTACCTCAGTTAACAATTTATA) were purchased from Gene Tools, LLC (Philomath, OR, USA).

### 2.9. Cell Migration Assay

Approximately 1.5 × 10^6^ cells were plated in 6-well plates and grown to 100% confluency. MLN4924 or DMSO was then added and the plate was scratched with a sterile pipette tip diametrically. To assess cell migration, the area of the scratch at the initial time (0) was compared with the area at the ending time (24 h). Several images were acquired using an inverted Olympus microscope.

### 2.10. Subcellular Fractionation

Transiently treated and/or transfected cells were washed 2× with PBS and then resuspended in cold hypotonic buffer containing 10 mM Tris-HCL (pH 7.5), 5 mM MgCl_2_, 1 mM DTT, and 1 mM PMSF. Cell lysis was performed by liquid nitrogen freezing and rapid thawing at 65 °C. Under these conditions, >95% of cells were lysed as monitored using microscopy. Samples were centrifuged at 600× *g* for 5 min at 4 °C to pellet nuclei and nuclei-associated structures (P1). To separate free cytoplasmic membranes (P100) and cytosolic proteins (S100), supernatants were ultracentrifuged (100,000× *g* for 45 min). Samples (P100) were resuspended in 2 × SDS loading buffer and analysed using Western blotting.

### 2.11. Immunoprecipitations

Cells were cultured in 10 cm dishes one day before reaching 85% confluency the next day. Cells were washed twice with cold PBS and scraped into 1 mL PBS. Cells were pelleted using centrifugation at 10,800 rpm for 20 s and resuspended in 550 µL NP-40 buffer (1% NP-40, 150 mM NaCl, 50 mM Tris-HCl Ph8.0, 5 mM EDTA pH8.0, 1 mM DTT), incubated for 30 min on ice before centrifugation at 13,000 rpm for 20 min. Protein concentration in the supernatant was determined using Bradford protein assay (Roche). Equal amount of protein was incubated with 1 µg of 4B2 anti-Mdm2 antibody at 4 °C for 3 h. In total, 20 µL of protein G sepharose (50% slurry, Sigma-Aldrich) pre-washed three times with NP-40 buffer were added and incubated at 4 °C for 1 h. Beads were washed 3 times in NP-40 buffer before elution in 50 µL 2×SDS loading buffer with boiling for 5 min. Eluates were analysed using Western Blot.

### 2.12. GST/GFP Pulldowns

**Expression and purification of GST fusion proteins.** pGEX plasmids expressing GST alone or GST-RhoA were transformed into Escherichia coli BL21 DE3. Bacteria were grown to an optical density at 600 nm (OD600) from 0.6 to 0.8 at 37 °C, and 0.2 mM isopropyl-β-d-thiogalactopyranoside (IPTG) was used for induction for 3 h at 37 °C with shaking. Bacterial pellets were resuspended in lysis buffer (50 mM Tris-HCl, pH 7.5, 50 mM NaCl, 5 mM MgCl_2_, 10µM GDP, 1 mM DTT) in the presence of protease inhibitors and lysed by sonication. 1% Triton X-100 was then added to enhance solubilization. Cell debris was removed by centrifugation for 15 min at 12,000× *g*, and Glutathione-Sepharose beads (GE Healthcare, Chicago, IL, USA) equilibrated in lysis buffer were added in the supernatant and incubated for 1 h at 4 °C. After washing the beads 3 times with ice-cold lysis buffer, protein concentration on beads was estimated using SDS-PAGE. The GST-RhoA (or GST alone) beads were used for in vitro binding assays, as described in [39].

For active RhoA pulldown assays, GST-RBD (expressing the RhoA binding domain of the RhoA effector Rhotekin) was produced similarly, except that IPTG induction was performed for 2 h, and lysis buffer did not contain GDP. For the experiment in Figure 4C, the fold activation for RhoA was calculated as follows: Signals were quantified with Image J, and for each condition, the RhoA-GTP/total RhoA ratio was used to determine the fold induction, with the sicntr untreated value as reference.

To generate the GDP- or GTPγS-loaded forms of RhoA, 10 μg recombinant GST-RhoA Sepharose beads were incubated with 0.5 mM GDP or GTPγS in 50 μL Nucleotide Exchange Buffer (NEB) (50 mM Tris-HCl, pH 7.5, 100 mM NaCl, 5 mM EDTA, 1 mM EGTA, 1 mM DTT and protease inhibitors) at 30 °C for 30 min. The loading reaction was stopped by adding MgCl_2_ to a final concentration of 20 mM. In total, 3–4 mg of cell lysates were subsequently incubated with 20 µg of RBD-GST Sepharose beads at 4 °C for 1 h. Eluates were obtained as described below.

### 2.13. Pull Downs

For GST/GFP pull downs, the cell pellets were lysed (50 mM Tris-HCl, pH 7.5, 150 mM NaCl, 1% Nonidet P-40, 10% Glycerol, 10 mM EDTA) with protease inhibitors cocktail (Roche) and incubated on ice for 10 min. The lysates were then syringed 15 times with a 21G needle and left on ice for 20 min, and then were centrifuged at 20,000× *g* at 4 °C for 15 min. Protein concentration in supernatants was measured with Bradford (Sigma-Aldrich) and equal amount of protein were used per condition. In total, 10 µL of the GST fusion proteins bound to glutathione-Sepharose beads: GST-RhoA, or GST alone were added to an equal amount of the supernatant and incubated overnight at 4 °C, shaking with 30 µL of prewashed Gluthation-Sepharose beads. For GFP pull downs, 20 μL of prewashed GFP-Trap^®^ (an immobilised single chain antibody, ChromoTek, Planegg, Germany) coupled to agarose were used. The next day, beads were collected using centrifugation (5000 rpm) for 1 min at 4 °C and then were washed three times with lysis buffer, and were eluted with 100 µL 2×SDS Laemmli buffer with 5% β-mercaptoethanol by boiling the samples for 5 min. Eluates were analysed using Western blotting.

### 2.14. Band Quantification

The intensity of bands from Western blot analysis was measured using ImageJ. The signal for each band was normalised with the signal of the corresponding loading control (β-actin, GAPDH, tubulin). Values represent the ratio of each condition over the control, used as reference.

## 3. Results

### 3.1. Prolonged Inhibition of the NEDD8 Pathway Targets Mdmx for Proteasomal Degradation

Previous studies have shown that NEDD8 inhibition by MLN4924 (Pevonedistat) activates the p53 pathway [40,41,42]. Several models have been proposed for p53 activation, including inhibition of CRL function, deNEDDylation of ribosomal proteins or of p53 itself [11,43,44,45]. During our experimentation with MLN4924, we found that inhibition of the NEDD8 pathway in A375 melanoma cells dramatically decreases the protein levels of Mdmx, but not of Mdm2 (Figure 1A). Consistent with previous studies, MLN4924 stabilises p53 [21,40,41,42].

In order to gain insights into the molecular mechanisms leading to the decrease in Mdmx protein levels upon NEDD8 inhibition, we determined the half-life of the Mdmx protein when protein NEDDylation is inhibited. For this, A375 cells were transfected with siRNA pools targeting either NEDD8 E2 conjugating enzymes, Ube2M (Ubc12) and Ube2F, and Mdmx half-life was determined by cycloheximide treatment. As shown in Figure 1B,C, knockdown of Ube2M but not of Ube2F dramatically reduces the half-life of Mdmx. Under these conditions, knockdown of Ube2M but not of Ube2F increased the levels of p53 (Appendix A). The data indicate that NEDD8 controls the stability of Mdmx post-transcriptionally specifically through Ube2M, but not Ube2F-dependent NEDDylation. Mdmx degradation induced by NEDD8 inhibition was partially rescued by the proteasome inhibitor MG132, indicating that NEDD8 controls Mdmx stability through the proteasome pathway (Figure 1D).

### 3.2. Inactivation of CRL4A and Mdm2 Are Required for Mdmx Degradation

Inactivation of CRLs is a key outcome of NEDD8 inhibition. To directly test the role of individual Cullin-based CRLs in Mdmx degradation, we transfected cells with a series of dominant negative (DN) Cullin mutants (Cullin1, 2, 3, 4A, 4B, 5) that inactivate the corresponding Cullin-based E3 ligases [46]. As shown in Figure 2A, expression of DNCullin4A and 4B resulted in a dramatic decrease in the levels of co-transfected HA-Mdmx, suggesting that the inactivation of CRL4A/4B targets Mdmx for degradation. To investigate whether the presence of Cullin4A/4B is required for Mdmx degradation upon inhibition of the NEDD8 pathway, we knocked-down Cullin4A or 4B with siRNAs and determined the effect on Mdmx protein levels upon MLN4924 treatment (Figure 2B). As expected in control siRNA transfected cells, MLN4924 decreased Mdmx levels. However, knockdown of Cullin4A but not 4B compromised the decrease in Mdmx levels upon MLN4924 treatment. The combination of the two experiments suggests that degradation of Mdmx upon inhibition of the NEDD8 pathway requires the presence of inactive Cullin4A. Due to the close homology between Cullin4A and B, it is likely that the decrease in Mdmx levels upon overexpression of the DNCullin4B is due to the indirect inactivation of Cullin4A.

Furthermore, we assessed the specificity of CRL4A in targeting Mdmx for degradation. For this purpose, we used low levels (5 nM) of ActinomycinD (ActD) that preferentially inhibit RNA Polymerase I, causing nucleolar stress. Under these conditions, ActD induces Mdmx degradation with no effect on Cullin NEDDylation [43,47]. We transfected A375 cells with either control or CRL4A siRNAs followed by Actinomycin D treatment (Figure 2C). CRL4A knockdown does not rescue Mdmx degradation induced by Actinomycin D, in contrast to what is observed with MLN4924 treatment. This underlines the specificity of the MLN4924-induced Mdmx degradation through inactivation of CRL4A and highlights the presence of multiple mechanisms for Mdmx degradation.

The Mdm2 E3-ligase is an established regulator of Mdmx stability [6,7]. Knockdown of Mdm2 by siRNA rescued the Mdmx degradation induced by MLN4924, indicating that Mdm2 is the E3 ligase involved in Mdmx degradation induced by NEDD8 inhibition (Figure 2D). The data suggest that Mdm2 and non-NEDDylated (inactive) Cullin4A are required for Mdmx degradation upon inhibition of NEDDylation. Previous studies indicated an interaction of Mdm2 with Cullin4A [48]. We therefore tested the possibility of the formation of a complex between Cullin4A-Mdm2-Mdmx and its regulation upon MLN4924 treatment. Immunoprecipitation experiments using anti-Mdm2 antibodies show that inhibition of NEDDylation promotes the interaction between Mdm2, non-NEDDylated (inactive) Cullin4A, and Mdmx (Figure 2E). This raises the possibility that Mdmx degradation occurs through the formation of a complex that includes inactive CRL4A and Mdm2, which acts as the active E3-ligase.

### 3.3. Mdmx Is Required for the Morphological Changes Induced in Metastatic Breast Cancer Cells upon NEDDylation Inhibition

A well-established effect of inhibition of the NEDD8 pathway is the induction of morphological changes in several types of cell lines through cytoskeletal reorganisation (see Introduction). Indeed, we found that, particularly in the metastatic breast cancer cells MDA-MB-231, MLN4924 treatment causes dramatic morphological changes, where cells adopt a rounded morphology compared to untreated elongated cells (Figure 3A, upper panels). We hypothesised that, should Mdmx degradation induced upon NEDD8 inhibition be required for the observed morphological changes, then Mdmx knockdown would enhance the effects of MLN4924 on cell morphology. Surprisingly, the opposite effect was observed. Knockdown of Mdmx almost completely blocked the morphological changes induced upon NEDD8 inhibition and cells retained the elongated morphology observed in control cells (Figure 3A, lower panels). While these data create a paradox, they nevertheless indicate a role of Mdmx in cell morphology regulation. The following experiments aimed to elucidate the potential role of Mdmx in cell morphology and to resolve the apparent paradox.

### 3.4. Mdmx Stabilises RhoA

Previous studies established that the stabilisation of the RhoA GTPase is critical for the induced morphological effects observed upon NEDD8 inhibition [24,36]. Indeed, in a time course experiment, MLN4924 treatment of A375 cells specifically stabilises RhoA, but not RhoC GTPase (Figure 3B). Additionally, under similar conditions, the levels of phosphorylated Myosin Light Chain (p-MLC2), a well-established phosphorylation target of the RhoA effector ROCK1/2, are also increased, confirming that NEDD8 inhibition causes RhoA stabilisation and activation (Figure 3C).

Active RhoA has a dual role in actin polymerisation and actomyosin contractility [49]. This is due to RhoA’s main downstream effectors: the actin nucleator mDia1 involved in actin filaments polymerisation, and the myosin 2 activator ROCK1/2 involved in actomyosin contractility [50]. RhoA induces contractile actin structures thanks to its two major effectors, the actin nucleator Dia1, and the myosin 2 activator ROCK1/2. While Dia1 can contribute to other structures, ROCK activity is central to actomyosin contractility [51]. We tested whether the ROCK pathway participates in the morphological changes induced by NEDD8 inhibition. Indeed, treatment of A375 and MDA-MB-231 cells with the ROCK1/2 inhibitor Y27632 completely abolished MLN4924-induced phosphorylation of MLC2. Importantly, this inhibitory effect on RhoA activation occurs despite RhoA stabilisation induced by MLN4924 (Figure 3D and Appendix A). These results indicate that NEDD8 inhibition promotes actomyosin contractility via the activation of the RhoA/ROCK pathway. Based on the observation that Mdmx knockdown prevented the MLN4924 induced rounded cell morphology, we tested whether Mdmx could control RhoA levels. Knockdown of Mdmx reduced the levels of RhoA and prevented RhoA stabilisation induced upon NEDDylation inhibition (Figure 3E,F). In addition, we performed half-life experiments to assess the effect of Mdmx knockdown on RhoA half-life. We employed conditions (high cell confluency) where Mdmx knockdown does not cause a dramatic decrease in RhoA levels in homeostatic conditions (Figure 4A, time point 0). Measurement of RhoA half-life shows that Mdmx knockdown reduces RhoA stability, suggesting that Mdmx protects RhoA from degradation (Figure 4A). Consistent with this hypothesis, we found that Mdmx knockdown increases the ubiquitination of wt GFP-RhoA (Figure 4B).

While the data show that Mdmx is required for the stabilisation of RhoA upon NEDD8 inhibition, kinetic analysis on the effect of MLN4924 on RhoA protein levels revealed a biphasic response. Initially, a gradual increase in RhoA levels is observed up to 24 h, followed by a decrease 72 h post-treatment (Figure 4C). Under these conditions, Mdmx levels were initially unchanged (8 h) and were maximally reduced at 72 h, correlated with the observed decrease in RhoA levels. Interestingly, under conditions where MLN4924 reduces the levels of RhoA (72 h treatment), we observed a modest stimulation of RhoA ubiquitination (Figure 4D). Thus, inhibition of the NEDD8 pathway initially causes the transient RhoA stabilisation that depends on Mdmx. Long-term inhibition of the NEDD8 pathway causes Mdmx degradation, which leads to the reduction in RhoA levels, possibly due to ubiquitin-mediated degradation. This is also consistent with the finding that Mdmx knockdown prevents RhoA stabilisation induced upon NEDD8 pathway inhibition.

### 3.5. Mdmx Is Required for RhoA Activation upon NEDDylation Inhibition

To further validate the role of Mdmx in RhoA activation, we used a biochemical approach, which is based on the ability of active RhoA (GTP-bound) to interact with the Rho-binding domain of Rhotekin (RBD), a RhoA effector. Extracts from control or Mdmx knockdown A375 cells either untreated or treated with MLN4924 were used in a GST-RBD pull-down before Western blotting against RhoA. The levels of active RhoA were markedly elevated in MLN4924 treated cells, but this effect was completely abrogated in Mdmx knockdown cells (Figure 5A). The combination of the above data supports a role for Mdmx in the control of RhoA stability and activity.

While RhoA in its GDP-bound form can shuttle between the cytosol and the plasma membrane, the association of RhoA with the plasma membrane is a step towards its activation by a specific GEF. It is expected that RhoA stabilisation will also result in higher RhoA levels at the plasma membrane, thus boosting its activation. Overactivation of RhoA and its target ROCK contributes to excessive stress fibers, leading to invasion inhibition [49,52,53]. Immunofluorescence analysis shows that NEDD8 inhibition promotes endogenous RhoA localisation to the plasma membrane, reminiscent of previously reported membrane blebs (Figure 5B) [54]. To further validate these findings, we performed subcellular fractionation in A375 cells, isolating cytoplasmic nuclear and membrane fractions. Mdmx knockdown prevented RhoA membrane localisation following NEDD8 inhibition, consistent with the imaging analysis for the requirement of Mdmx for RhoA activation (Figure 5B,C).

Additionally, we monitored actin stress fiber formation by phalloidin staining. While control cells show little or no stress fibers, NEDD8 inhibition provokes a robust stress fiber formation (Figure 5D). Critically, Mdmx knockdown almost completely blocked the stress fiber formation induced by MLN4924 (Figure 5D). Collectively, the above data show that Mdmx is required for RhoA stabilisation, increased membrane association, and activation through membrane targeting induced by NEDD8 inhibition.

### 3.6. Mdmx Preferentially Interacts with the Inactive State of RhoA through Its Zinc Finger Domain

To provide further insights into the functional link between Mdmx and RhoA, we determined whether Mdmx interacts with RhoA. We used constructs expressing wild type GFP-RhoA or RhoA mutants that represent either inactive (GFP-RhoAN19) or constitutively active RhoA (GFP-RhoAV14). These constructs were transfected in A375 cells alone or co-transfected with HA tagged Mdmx, followed by a GFP-trap pull-down and Western blot analysis. As shown in Figure 6A, both endogenous and overexpressed Mdmx preferentially bind to the inactive but not to the active form of RhoA. Mdmx also interacts with wild type RhoA, but to a smaller extent than RhoAN19.

We further confirmed the above observations using an alternative biochemical approach. We used recombinant GST-RhoA loaded either with GDP or GTPγS and immobilised on glutathione-agarose beads [39]. Beads were incubated with A375 cell extracts, and endogenous Mdmx binding was monitored by Western blotting. Consistent with the above data, Mdmx preferentially interacted with GDP-bound RhoA (Figure 6B).

To map the binding domain of Mdmx on RhoA, we tested the interaction between a series of Mdmx mutants and the RhoAN19 mutant, which showed the highest binding to Mdmx. We used constructs that either express or do not express the N-terminal p53 binding domain of Mdmx, the triple phosphorylation Mdmx mutant S342/S367/S403A (3SA) that controls Mdmx stability upon DNA damage and constructs with the Zn finger domain deleted [55] (Figure 6C). With the exception of the N-terminal Mdmx fragment (1–153), all of the tested mutants were able to interact with RhoAN19, indicating that the Mdmx-RhoA interaction is independent of p53 binding and the C-terminal phosphorylation sites of Mdmx (Figure 6C,D). In contrast, the interaction of RhoAN19 with the Mdmx mutant lacking the Zn finger domain (Mdmx Δ300–328) was severely compromised compared to the wild type (Figure 6E).
Figure 6Mdmx interacts with the inactive GDP form of RhoA through its Zn finger domain. (**A**) A375 cells were transfected with GFP-tagged RhoA constructs (3 μg) and HA-Mdmx (3 μg) as indicated. After 24 h post-transfection, extracts were used for GFP trap pull-downs (PD) before eluates, and total cell extracts were used for Western blot analysis. SE: Short exposure, LE: Long exposure. (**B**) Recombinant GST-RhoA loaded in vitro either with GDP or GTPγS was used for pull-down assays with A375 cell extracts. Eluates and total cell extracts were used for Western blot analysis. (**C**) Schematic representation of the Mdmx protein domains. ZF: Zinc Finger, RF: Ring Finger, NoLS: nucleolus localisation signal-adapted from [56]. The * indicate the phosphorylation sites S342/S367/S403 mutated into Alanine (3SA Mdmx mutant). (**D**) Experiment was performed as in (**A**) using, instead, the indicated Mdmx constructs and the GFP-RhoAN19 mutant. GFP trap pull-down (PD) eluates and total cell extracts were used for Western blotting. (**E**) Experiment as in (**D**) using the wild type and Zn finger (ZF) deletion Mdmx mutant. Arrows indicate the wild type and Zn finger deletion mutant.
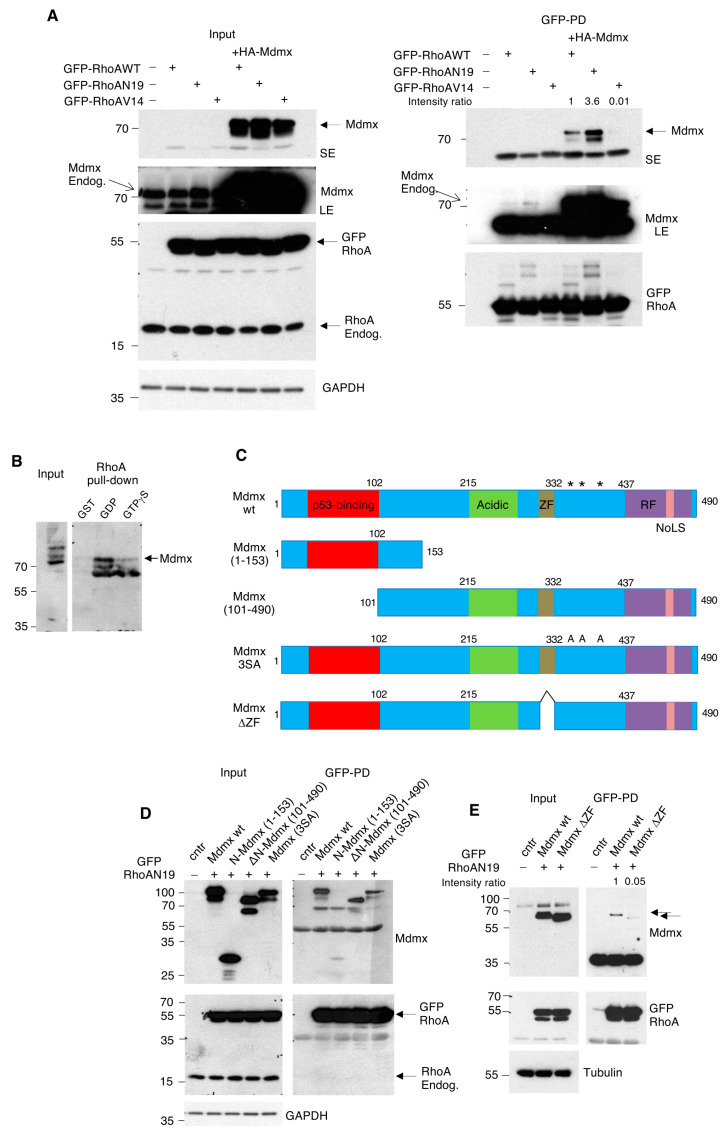


### 3.7. NEDDylation Inhibition Impairs Xenopus Gastrulation in a Mdmx and RhoA-Dependent Manner

To further address the physiological relevance of this new link between the NEDD8 pathway, Mdmx, and RhoA in an in vivo system, we turned to the Xenopus embryo model. Xenopus gastrulation involves massive tissue rearrangement, in particular, the internalisation of the mesoderm, which is a major drive of this global reorganisation. During this process, mesoderm cells acquire high collective migration capabilities, requiring tissue “softening”, which is mainly achieved through developmentally regulated expression of inhibitors of the RhoA-ROCK-myosin II pathway [57]. Experimental interference with these RhoA inhibitors that results in RhoA activation severely impacts mesoderm internalisation and compromises gastrulation [57].

We tested the effect of MLN4924 treatment on Xenopus embryo development. The progress of gastrulation can be conveniently monitored by the appearance of the so-called blastopore, delimited by a typical “blastopore lip”, and its subsequent progressive closure (Figure 7A). MLN4924-treated embryos underwent normal early development, including cleavage and formation of the blastula, but showed defective gastrulation, as unambiguously reflected by impaired closure of the blastopore, reminiscent of what is observed upon depletion of RhoA-ROCK inhibitors [57] (Figure 7B). The effect was quantified by comparing the blastopore area at midgastrula stage (Figure 7C). Phenotypes ranged from slightly delayed closure to fully open blastopores, (Figure 7B, red arrows). In the latter cases, embryos died during or shortly after gastrulation. In the milder cases, embryos continued development past neurulation, but showed severe defects, particularly in head structures, which is the characteristic phenotype resulting from partially impaired mesoderm internalisation (Figure 7D,E).

We tested the role of Mdmx in the MLN4924 response by monitoring the effect of MLN4924 treatment in control embryos or embryos injected with an antisense morpholino oligonucleotide specifically targeting Mdmx (Mdmx MO). We found that while Mdmx MO had no significant effect on gastrulation in DMSO-treated embryos, it fully rescued the defects observed upon MLN4924 treatment (Figure 7B,C). Gastrulation was also fully rescued by injecting low amounts of mRNA coding for a dominant negative RhoA variant (Figure 7B,C), consistent with the idea that the gastrulation defects induced upon MLN4924 treatment are RhoA dependent.

### 3.8. Mdmx Is Required for the Anti-Invasive Effects of NEDDylation Inhibition

The in vitro and in vivo data reveals that NEDD8 inhibition causes cell morphological changes and defects in Xenopus embryo gastrulation through an Mdmx-dependent RhoA activation. These phenotypes are associated with the resulting increased cell contractility, a process also required for the anti-invasive effect of NEDD8 inhibition in metastatic tumour cells [24]. Based on the functional relationship between Mdmx and RhoA activation, we tested the role of Mdmx in cell-invasion. For this, we employed a 3D collagen matrix gel assay, which reconstitutes in vitro the ability of cells to invade through a collagen matrix. A375 and MDA-MB-231 cells were treated with DMSO or MLN4924 only during the invasion assay for 24 h. The doses of MLN4924 used (50–200 nM) for the period of the experiment, while they block Cullin NEDDylation and stabilise RhoA, do not cause severe defects in the cell cycle or induce apoptosis (Appendix A). Consistent with previous reports, MLN4924 dramatically reduced cell invasion in both tested cell lines [22,23,24,26,27] (Figure 8A). We also assessed the role of NEDD8 in cell migration in a 2D environment by performing a wound healing experiment, which, in contrast to the 3D Matrix experiment where RhoA was the key regulator, is mainly controlled by activation of the Rac1 GTPase activation at the leading edge [58]. No change in the wound closure was detected upon MLN4924 treatment in both A375 and MDA-MB-231 cell lines, indicating that NEDD8 does not affect Rac1-dependent cell migration in a 2D environment (Figure 8B and Appendix A). Consistent with this, the levels of Rac1 remain unaffected upon MLN4924 treatment or Mdmx knockdown compared to the control cells (Figure 8C). These results lead us to conclude that inhibition of the NEDD8 pathway compromises cell invasion but not Rac1-dependent migration. Focusing on the role of NEDD8 on cell invasion, we found that Mdmx knockdown compromised the anti-invasive effect of NEDD8 inhibition (Figure 8D). Therefore, Mdmx mediates the effect of NEDD8 inhibition and prevents cell invasion potentially through regulation of RhoA stability and activity.

## 4. Discussion

### 4.1. Mdmx Is Required for RhoA Stabilisation and Activation Induced by NEDDylation Inhibition

The analysis of the effect of NEDD8 inhibition on the expression levels of key regulatory components of the p53 pathway and cell morphology revealed an intriguing role for Mdmx in cell invasion through the RhoA GTPase. The NEDD8 pathway controls the stability of RhoA through activation of the SCF^FBXL19^ and Cullin3^BACURD^ CRLs that preferentially degrades the GDP-bound state of RhoA [36,59]. Upon NEDD8 inhibition and CRL inactivation, RhoA is stabilised and subsequently activated [36]. The induced stabilisation/activation of RhoA is regarded as a key molecular event for the anti-invasive effects of NEDD8 inhibitors [24]. Our studies revealed that Mdmx is directly involved in this process. By preferentially interacting with the GDP-bound state, Mdmx protects RhoA from proteasomal degradation and participates to the subsequent activation of RhoA. A series of biological and biochemical assays that are regarded as hallmarks for monitoring RhoA activation, including loading of RhoA with GTP, translocation of RhoA to the plasma membrane, activation of well-established downstream RhoA pathway components such as MLC2, and the formation of excessive stress fibers, show that Mdmx is a critical required element of RhoA stabilisation and subsequent activation upon NEDD8 inhibition. These effects are reminiscent of the proposed role of the GDIs as molecular chaperones that protect the inactive GDP-loaded Rho GTPases from degradation and facilitate GTPase membrane association and activation [60,61]. Our working model is that by stabilising the GDP loaded form of RhoA, Mdmx maintains the required RhoA pool and promotes the association of RhoA to the plasma membrane for subsequent activation by its specific RhoGEF.

### 4.2. Regulation of Mdmx Stability by the NEDD8 Pathway: Remodelling the CRLs to Degrade Mdmx?

The above-described role of Mdmx in RhoA stabilisation and activation is dependent on the extent of NEDD8 inhibition. This is because Mdmx is also under the control of the NEDD8 pathway. Transient inhibition of NEDDylation stabilises RhoA in an Mdmx-dependent manner as the first step of the response. However, prolonged inhibition of NEDDylation leads to Mdmx degradation and subsequently to reduced protein levels of RhoA (Figure 8E). The data indicate that inhibition of the NEDD8 pathway promotes the complex formation between inactive (non-NEDDylated) Cullin4A, Mdm2, and Mdmx. Mdm2 is an established Ring E3-ligase for Mdmx, required for Mdmx degradation induced upon NEDD8 inhibition. The interaction of Mdm2 with Cullin4A was previously reported and, similarly to our findings, it was indicated that Mdm2 interacts with the non-NEDDylated (inactive) form of Cullin4A [48]. The studies support the idea that while inhibition of the NEDD8 pathway blocks CRL activity, causing the rapid accumulation of canonical CRL substrates such as RhoA, it also promotes the formation of new complexes between non-NEDDylated Cullins and Ring E3-ligases that can promote the degradation of non-canonical CRL substrates (neosubstrates) such as Mdmx (Figure 8E). Similar conditions that remodel the activity of CRLs have been reported for the action of immunomodulatory drugs such as thalidomide, which upon binding to CRL4^CRBN^ alter the E3-ligase specificity towards non-canonical substrates [62,63,64]. Intriguingly, in the case of RhoA and Mdmx, the opposing effects of CRL inhibition on their stability are functionally linked, as the induced RhoA stabilisation strictly depends on Mdmx (Figure 8E).

### 4.3. An Anti-Invasive Function for Mdmx

The role of RhoA in cell migration and invasion has been an intense area of research, which revealed diverse and often opposing functions for RhoA [49]. Depending on the activation of downstream RhoA effectors and/or the activation of additional Rho GTPases including Rac1 and Cdc42, RhoA can either promote or inhibit cell invasion. The RhoA-mediated stimulation of cell protrusions at the leading edge of migrating cells promotes cell invasion, whereas RhoA-mediated induction of stress fiber formation that enhances actomyosin contractility and cell stiffness reduce the invasive potential [65]. The above-described opposing activities of RhoA are coordinated, supporting the concept that a decrease in RhoA-mediated contractility is required to promote the formation of membrane protrusions and to increase the cell invasion potential [66].

Our in vitro and in vivo data support the concept that the induced activation of RhoA upon NEDDylation inhibition increases cell contractility that promotes the observed morphological changes and defects in embryo gastrulation. Subsequently, the increase in RhoA activation reduces the invasive potential of metastatic cell lines. Mdmx is a key factor in this process by specifically stabilising the GDP state of RhoA, but not affecting the stability of additional Rho GTPases, including Rac1 and RhoC, that are implicated in cell motility. While Mdmx is also required for RhoA stabilisation under homeostatic conditions (Figure 3), the biological significance of this role of Mdmx may be more relevant in conditions where RhoA is overactivated. In addition to the presented MLN4924-induced RhoA overactivation in cancer cell lines, Mdmx may be involved in conditions such as hypertension or decreased brain size due to Cul3 mutations where RhoA is also found to be overactivated [67,68].

### 4.4. Mdmx as Tumour Suppressor through the Zinc Finger Domain

Recent studies suggested a paradoxical tumour-suppressor function of Mdmx, which can be either p53-dependent or -independent. In particular, in p53−/− mouse models, Mdmx acts as tumour suppressor, preventing cell invasion and chromosomal instability [69]. These tumour-suppressive functions of Mdmx were mapped within the Zn finger domain, which we identify as the domain required for the binding of Mdmx to RhoA [10]. Critically, the findings that the Mdmx-RhoA interaction is independent of the Mdmx-p53 binding domain and that the effects of Mdmx on cell invasion are observed both in p53 wild type (A375) and p53 mutant cells (MDA-MB-231), possibly indicating that Mdmx does not require p53 for the control of these processes.

The data on the anti-invasive role of Mdmx through control of RhoA now propose a molecular pathway for the reported tumour-suppressive functions of Mdmx. Deregulation of Mdmx protein levels due to amplification of the Mdmx gene and its splice variants is observed in a broad spectrum of tumours and is often related to poor prognosis [70]. Collectively, the presented data suggest that the levels of Mdmx in tumours may be a critical determinant for the efficacy of NEDD8 inhibitors in the clinic through control of RhoA stability/activity. The so-called Maximum Tolerated Dosage and the timing and intervals of drug administration are critical parameters for the efficacy of any chemotherapy [71]. The ongoing clinical trials have indicated that MLN4924/Pevonedistat is generally well tolerated with minimal toxicity [11]. However, the finding that prolonged inhibition of NEDDylation reduces RhoA levels due to the induced degradation of Mdmx indicates that long-term treatments may compromise the anti-invasive effects of NEDD8 inhibitors. Assessment of this hypothesis in vivo may have significant impact on the administration protocols for NEDD8 inhibitors in the clinic.

## 5. Conclusions

In addition to its well-established function as an oncogene through the inhibition of the p53 tumour suppressor, Mdmx has also been reported to repress transformation and tumour progression. However, the molecular basis for this potential tumour-suppression role of Mdmx remains unknown. We found an anti-invasive role for Mdmx through control of the RhoA GTPase stability. Mdmx directly binds to the inactive GDP-form of RhoA and protects RhoA from degradation. This is a critical step for RhoA stabilisation upon inhibition of the ubiquitin-like molecule NEDD8 pathway. By stabilising RhoA, Mdmx mediates the anti-invasive effects of NEDD8 inhibitors. This study reveals the RhoA signalling module to be a molecular pathway through which Mdmx mediates its reported anti-oncogenic functions. As inhibitors of the NEDD8 pathway are in clinical trials, the status of Mdmx may be a critical determinant for the anti-tumour effects of these inhibitors.

## Figures and Tables

**Figure 1 cells-13-01625-f001:**
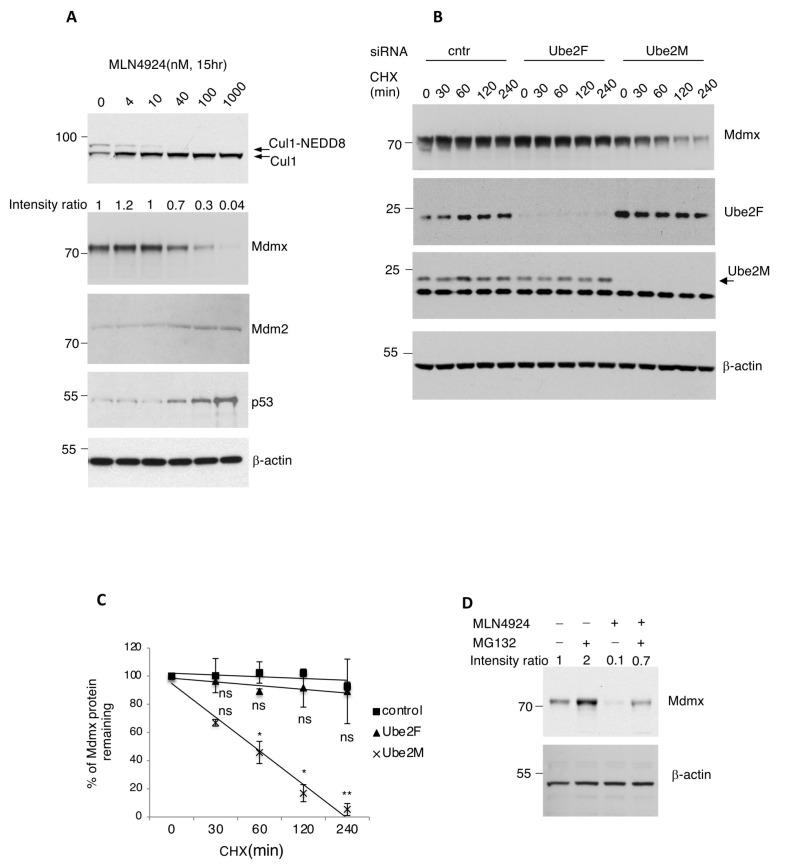
Prolonged inhibition of the NEDD8 pathway promotes Mdmx degradation. (**A**) A375 melanomas cells were treated with increasing concentrations of the NEDD8 inhibitor MLN4924, and total cell extracts were analysed using Western blotting for the indicated proteins. (**B**) Half-life experiment for Mdmx in A375 cells upon knockdown of the NEDD8 E2-conjugating enzymes Ube2F and Ube2M using cycloheximide (30 μg/mL). (**C**) Quantification for the experiment presented in B. Values represent the mean of three independent experiments ± SD. *p*-values * < 0.05, ** < 0.01 by Student’s *t*-test. ns: non-significant. (**D**) A375 cells were treated with MLN4924 (1 μM, 15 h) and MG132 (30 μM, 4 h), and cell extracts were used for Western blotting.

**Figure 2 cells-13-01625-f002:**
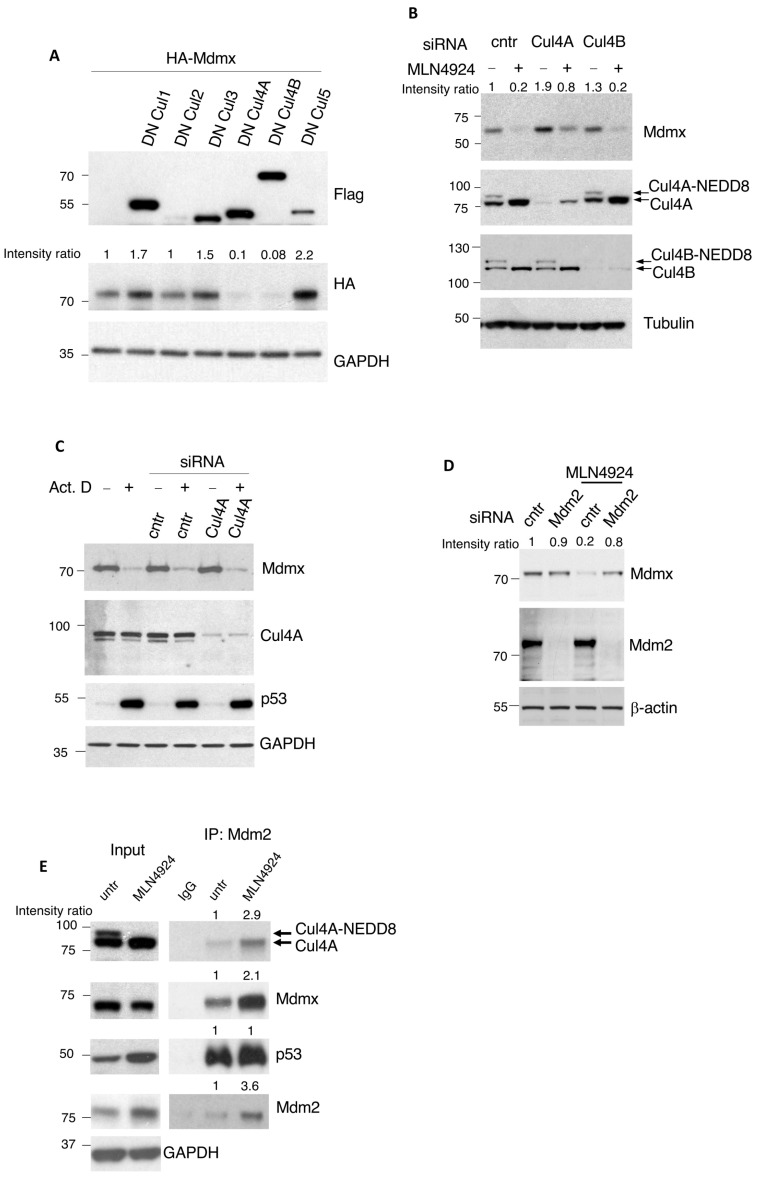
Mdmx degradation induced upon NEDDylation inhibition requires inactive Cullin4A and Mdm2. (**A**) A375 cells were co-transfected with different dominant negative-expressing Flag-tagged Cullin constructs (3 μg) and HA-Mdmx (3 μg). After 48 h post-transfection, total cell extracts were used for Western blotting as indicated. (**B**) A375 cells were transfected with either control, Cullin4A, or Cullin4B siRNAs followed by MLN4924 treatment (1 μM, 15 h). After 48 h post-transfection, cell extracts were used for Western blotting against the indicated proteins. (**C**) Experiment performed as in B using Actinomycin D treatment (5 nM, 15 h). (**D**) A375 cells were transfected with Mdm2-targeting siRNAs and treated as indicated (MLN4924, 1 μM, 15 h), before extracts were analysed by Western blotting using indicated antibodies. (**E**) A375 cells were treated with MLN4924 (1 μM, 6 h, conditions where Mdmx levels are not significantly affected) and extracts were used for immunoprecipitations using 4B2 anti-Mdm2 antibody. Western blot analysis on immunoprecipitates and total cell extracts was performed with the indicated antibodies.

**Figure 3 cells-13-01625-f003:**
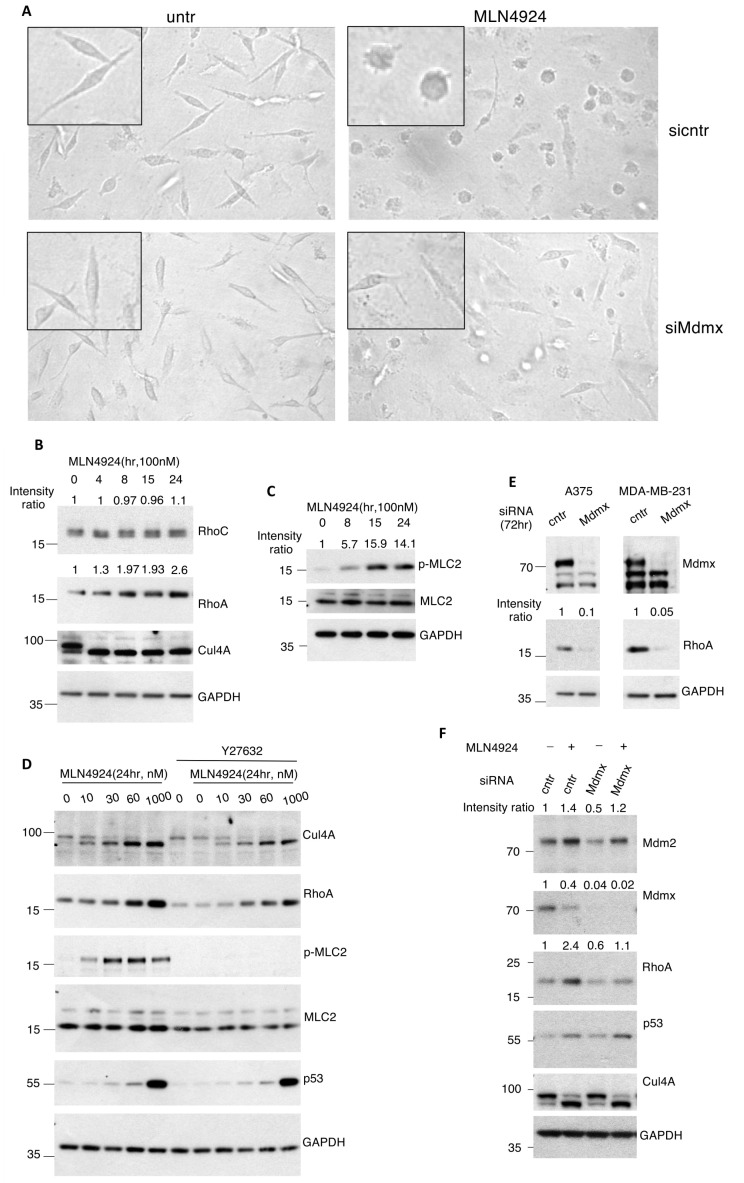
Mdmx stabilises RhoA and is required for the morphological changes induced upon NEDDylation inhibition in metastatic cell lines. (**A**) MDA-MB-231 cells were transfected with either control or Mdmx siRNAs before treatment with MLN4924 (0.5 μM). Cell morphology was monitored 24 h post-treatment. (**B**,**C**). A375 cells were treated as indicated, and extracts were used for Western blotting using indicated antibodies. (**D**) A375 cells were treated with MLN4924 and the RhoA/ROCK inhibitor Y27632 (10 μM, 2 h) as indicated, and total cell extracts were used for Western blotting using indicated antibodies. (**E**) A375 and MDA-MB-231 cells were transfected with Mdmx siRNAs for 72 h before cells were harvested, and extracts were used for Western blotting using indicated antibodies. (**F**) A375 cells were transfected with control or Mdmx siRNAs and treated with MLN4924 (0.5 μM) 24 h before harvesting (48 h in total). Western blotting on cell extracts was performed as indicated.

**Figure 4 cells-13-01625-f004:**
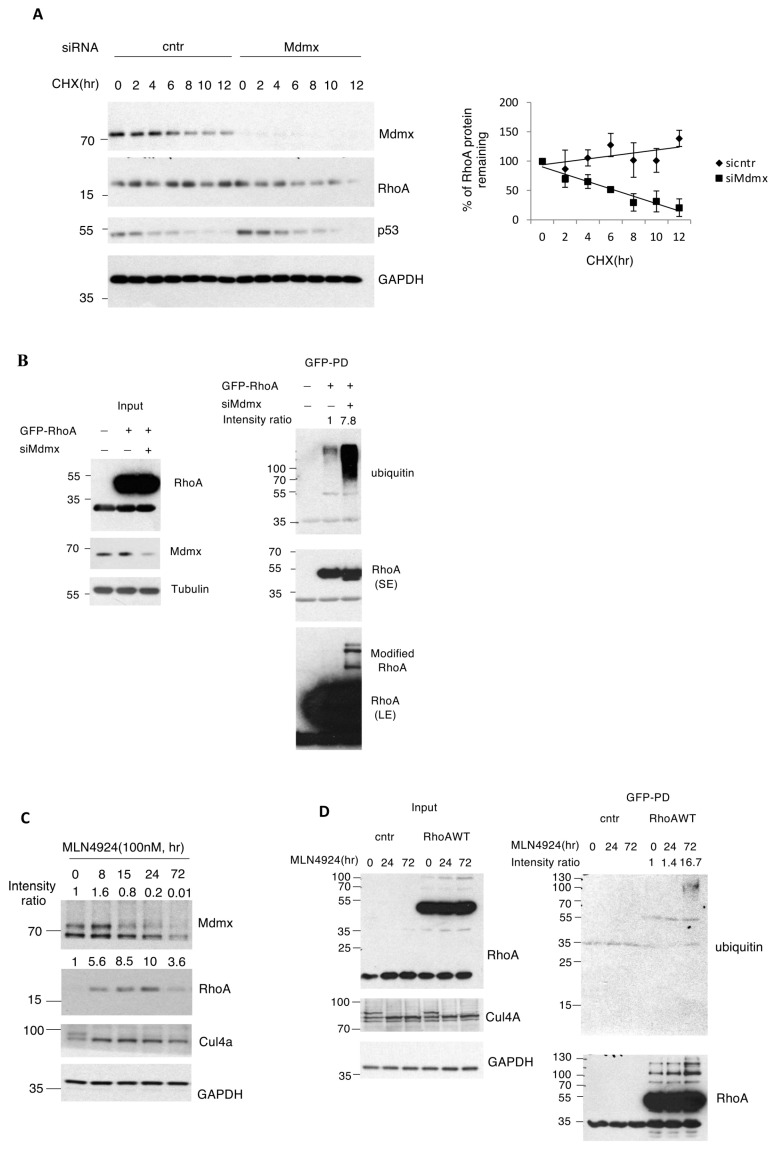
Mdmx is required for RhoA stabilisation and activation induced upon NEDDylation inhibition. (**A**) A375 cells transfected with control or Mdmx siRNAs for 72 h were then treated with cycloheximide (30 μg/mL), cells were harvested at the indicated times before Western blot analysis of cell extracts. Right graph represents the quantification of RhoA levels in the experiment. Values represent the mean of three independent experiments ± SD. (**B**) A375 cells transfected with control or Mdmx siRNAs were then transfected with control empty vector or wild type GFP-RhoA constructs as indicated. To prevent RhoA degradation, proteasome inhibitors (MG132, 30 μM, 5 h before harvesting) were used. Upon GFP-RhoA pull-downs (PD) the eluates were blotted with anti-ubiquitin antibodies. SE: Short exposure, LE: Long exposure. (**C**) A375 cells were treated with MLN4924 (100 nM) for the indicated times, and cell lysates were analysed using Western blotting as indicated. (**D**) A375 cells were transfected with either control or GFP-tagged wild type RhoA and treated with MLN4924 (100 nM) for the indicated times. To prevent RhoA degradation, proteasome inhibitors (MG132, 30 μM, 5 h before harvesting) were used for all indicated time points. GFP trap pull-down (right panel) was performed, and RhoA ubiquitination was analysed using Western blotting.

**Figure 5 cells-13-01625-f005:**
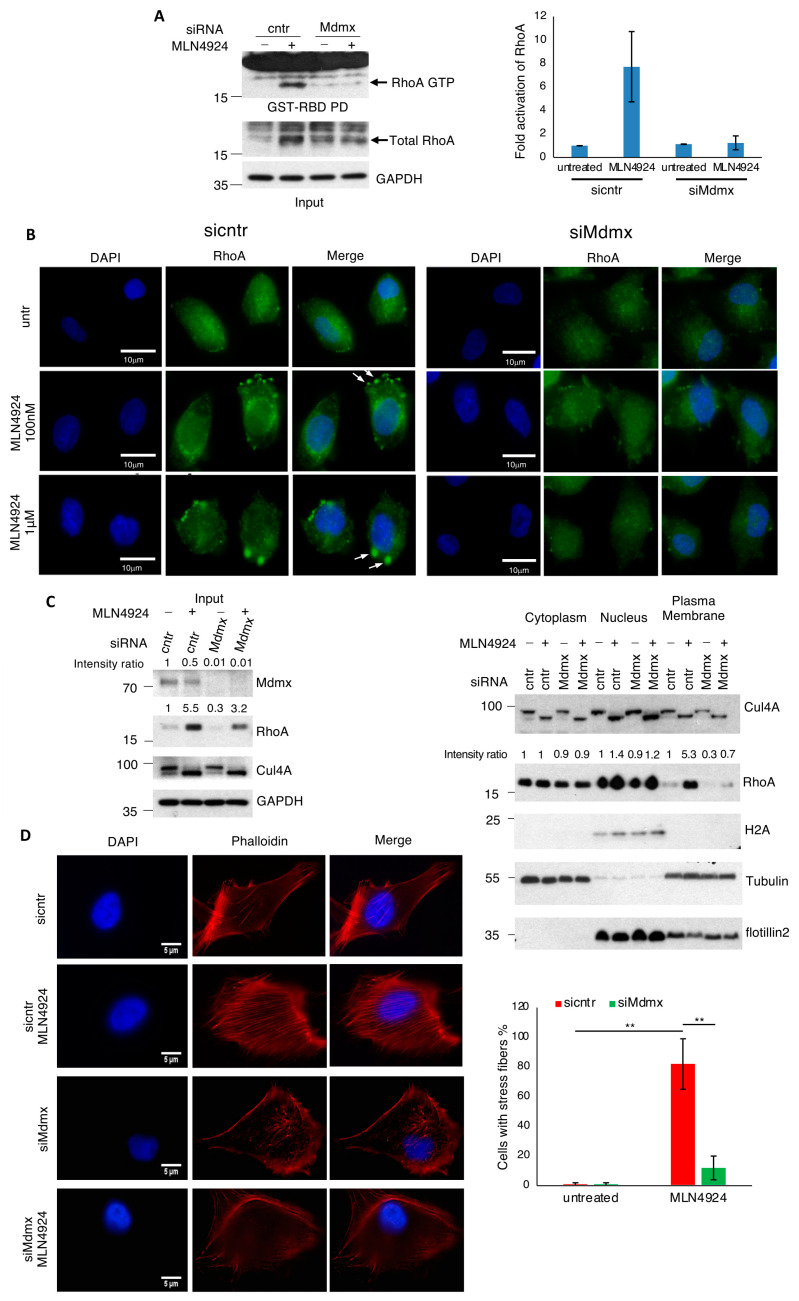
Mdmx is required for RhoA activation at the plasma membrane. (**A**) A375 cells transfected with control or Mdmx siRNAs were treated with MLN4924 (1 μM, 15 h), and extracts were used in a GST-Rhotekin pull-down to isolate the active GTP-loaded state of RhoA. Eluates and total cell extracts were used for Western blotting. The graph represents the mean of two independent experiments ± SD of the fold activation of RhoA (Materials and Methods). (**B**) A375 cells transfected with control or Mdmx siRNAs were treated with MLN4924 (15 h) before the analysis of RhoA using immunofluorescence. Arrows indicate the localisation of RhoA in the plasma membrane upon MLN4924 treatment. (**C**) Similar experiment as in A (MLN4924, 1 μM, 15 h), but cells were used for the isolation of plasma membranes. Total cell extracts (**left panel**) and isolated fractions (**right panel**) were used for Western blot analysis. (**D**) Experiment performed as in (**A**), using low doses of MLN4924 (100 nM, 15 h) to prevent the induced morphological changes, and cells were used for Phalloidin staining to monitor stress fiber formation. Right panel is the quantification of the experiment performed in (**D**), using 100 cells/condition (*n* = 3), ±SD. *p*-values, ** < 0.01.

**Figure 7 cells-13-01625-f007:**
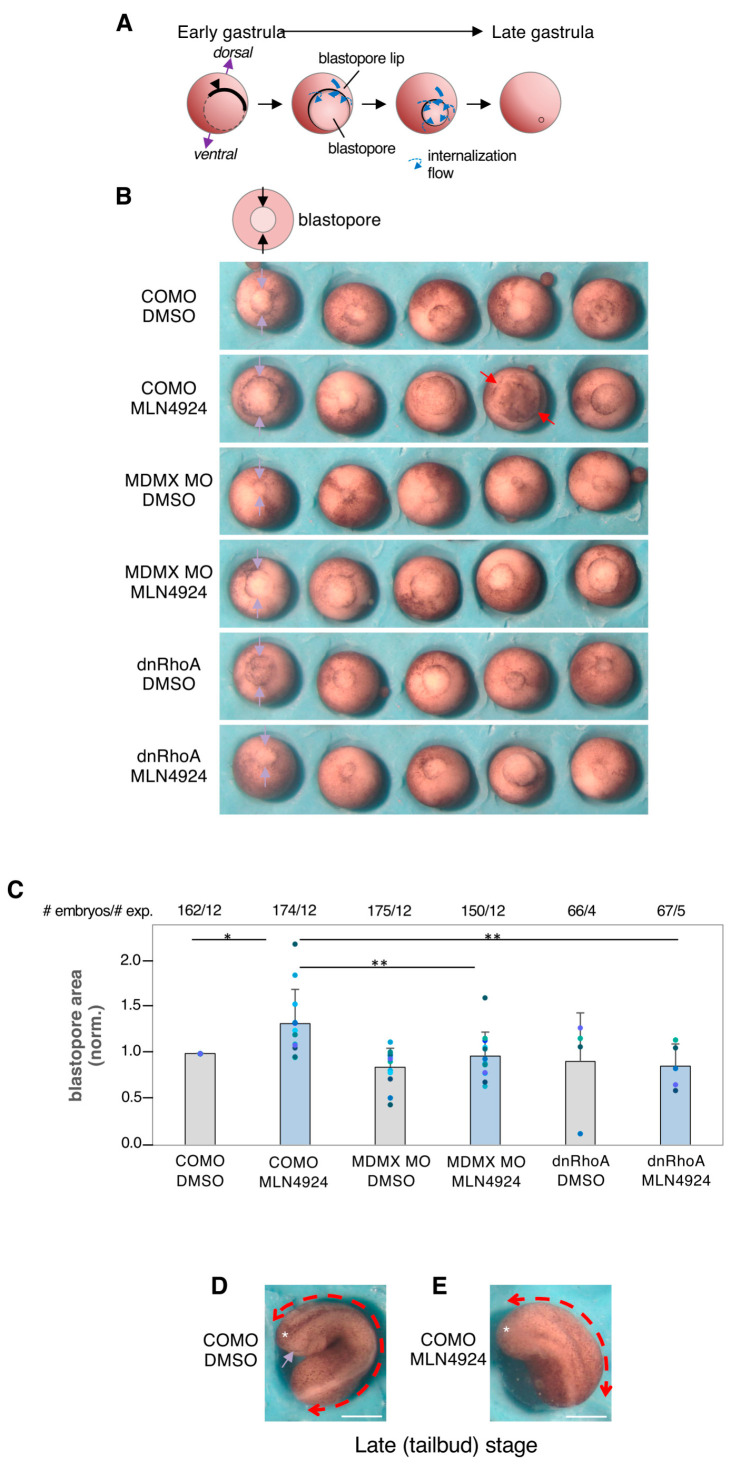
Mdmx and RhoA-dependent effect of MLN4924 treatment on Xenopus gastrulation. (**A**) Schemes of Xenopus embryo external morphology at four consecutive stages of gastrulation. The onset gastrulation is marked by the appearance of pigmented cells that outline the future blastopore (arrowhead). A flow of mesoderm internalisation is established (blue arrows), starting on the dorsal side and propagating all around to form the so-called blastopore lip. As internalisation progresses, the blastopore progressively shrinks. Its complete closure marks the end of gastrulation. (**B**) Representative images of five embryos per condition, from the same experiment, imaged at the mid–late blastula (stage 12). The embryos were oriented bottom-up to view the blastopore (arrows). Embryos were injected at the two cell stage with control morpholino (COMO), Mdmx-specific MO (MdmxMO), or mRNA coding for dominant negative RhoA N19 (dnRhoA). From blastula stage on, embryos were incubated in the presence of 20 μM MLN4924. Solvent DMSO (1/500) was used as negative control. Scale bar, 500 μm. (**C**) Quantification of relative blastopore area, normalised for each experiment to the average area of COMO-DMSO controls. Total number of embryos and number of independent experiments are indicated at the top of the graph. The coloured dots indicate the mean values for each experiment. Statistical comparison: non-parametric Anova (Kruskal–Wallis) followed by post hoc Dunn’s test. *p*-values, * < 0.05, ** < 0.01. (**D**,**E**) Examples of control and MLN4924-treated embryos at a late stage (tailbud). (**D**) The embryo has become thin and elongated (red double arrowhead). It is curved because still confined by the transparent egg shell. The anterior part (asterisk) shows well-defined head structures (purple arrow, optic anlage). (**E**) MLN4924-treated embryos show a typical phenotype resulting from moderately defective gastrulation. The embryo axis has remained much shorter, and the head structures are missing. Scale bar, 500 μm.

**Figure 8 cells-13-01625-f008:**
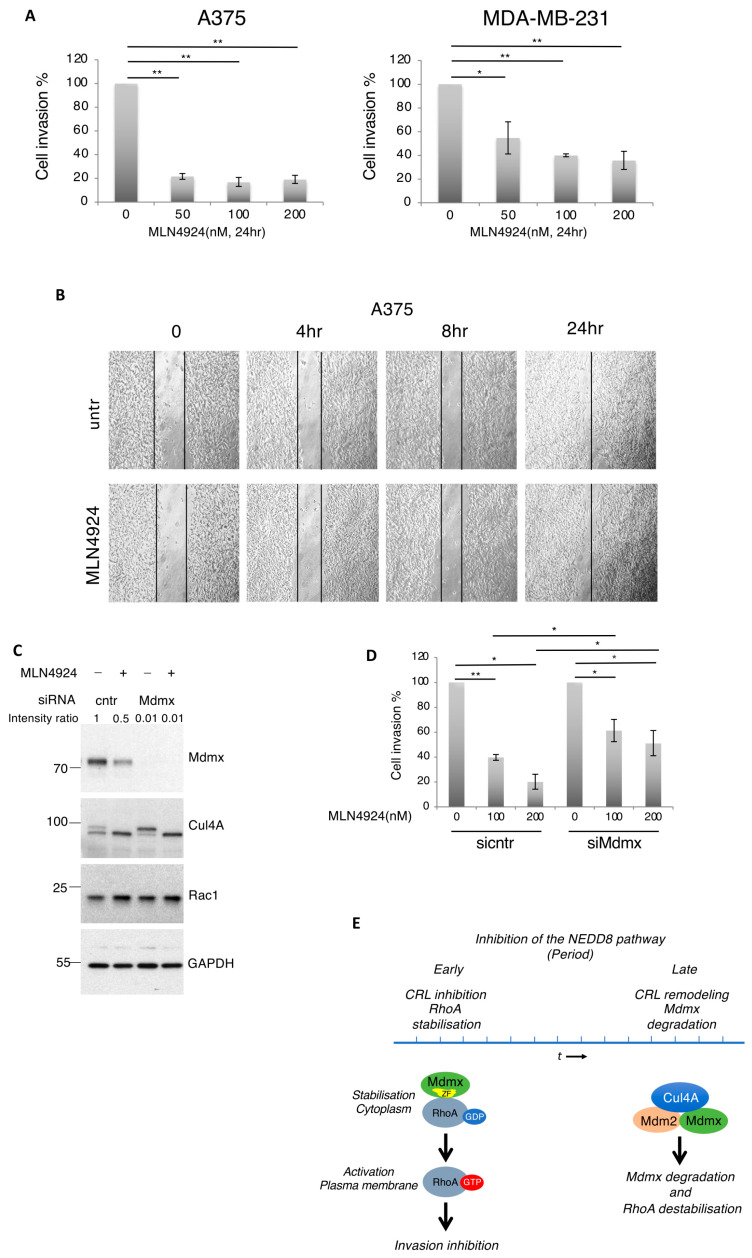
Mdmx prevents cell invasion but not 2D cell migration of metastatic cells. (**A**) A375 and MDA-MB-231 cells were used in a 3D cell invasion assay and the effect of NEDD8 inhibition was tested as described in Materials and Methods. Values represent the mean of 4 independent experiments ± SD. *p*-values * < 0.05, ** < 0.01 by Student *t*-test. (**B**) Similar experiment was performed to test the effect of NEDD8 inhibition (MLN4924, 1 μM) on 2D cell migration. Black lines indicate the gap between cells filled over the indicated period of time. (**C**) A375 cells transfected with control or Mdmx siRNAs were treated with MLN4924 (1 μM, 15 h) as indicated and cell extracts were used for Western blot analysis. (**D**) Similar experiment as in A, but 36 h prior to MLN4924 treatment, A375 cells were transfected with control or Mdmx siRNAs. Values represent the mean of four independent experiments ± SD. *p*-values * < 0.05, ** < 0.01 using Student *t*-test. (**E**) Transient inhibition of NEDDylation and CRL inactivation results in RhoA stabilisation and activation that depends on Mdmx. Prolonged NEDD8 inhibition allows for the formation of complexes between non-NEDDylated Cullin4A (inactivation of CRL4A) and E3-ligases (Mdm2) that promote the degradation of Mdmx and subsequent destabilisation of RhoA.

## Data Availability

Any requests regarding presented data and/or reagents should be addressed to Dimitris Xirodimas, dimitris.xirodimas@crbm.cnrs.fr.

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
