# Peer review of "An Anti-Invasive Role for Mdmx through the RhoA GTPase under the Control of the NEDD8 Pathway"

_cells, 2024, doi:10.3390/cells13191625_

Round 1

Reviewer 1 Report

Comments and Suggestions for Authors

Journal of cells

Research Article.

The article entitled An anti-invasive role for Mdmx through the RhoA GTPase un-2 der the control of the NEDD8 pathway’’. Mdmx (Mdm4) is recognized as an oncogene primarily via suppressing the p53 tumor suppressor. The impact of NEDD8 pathway inhibitors on p53 activation, human cell morphology, and cell motility during gastrulation in Xenopus embryos demonstrated that Mdmx has an anti-invasive role. Mdmx plays a crucial role in the prevention of invasion by NEDDylation inhibitors by stabilizing and activating the RhoA GTPase. The degradation of Mdmx necessitates the presence of Culin4A which is non-NEDDylated (inactive) and the Mdm2 E3-ligase. The authors demonstrate that Mdmx can regulate cell invasion by stabilizing and activating RhoA, which may be connected to the claimed anti-cancer properties of Mdmx. The state of Mdmx may play a crucial role in determining the effectiveness of these inhibitors against tumors.
After a thorough examination of the work, I determined that it is appropriate for publishing in the journal. I thus acknowledge the acceptance of this
manuscript for potential publication, contingent upon the implementation of significant revisions as well as minor adjustments. The article has many prevalent errors that the writers should rectify. Once all the errors have been rectified, the work may be deemed suitable for publishing in the esteemed Cells Journal.

Comments for Authors

Major changes must be included.

Ø  It would be better to include the quantification of WB in the figure in the entire manuscript. And also make the figure in consistent distance and size.

Ø  Section Introduction: Revise it and make it clear.

Ø  The author discussed “previous studies. Why didn’t the author give the reference and need to revise and discuss the previous study and explain the previous study and this study’s result?

Ø  Figure 4 and figure 6 WB results need to be repeated. The author needs to crop or fix the WB result.

Mainor changes

Ø  Write keywords in alphabetical order.

Ø  The authors want to put more latest related citations in the introduction part.

Ø  In (4.7 Cell Invasion assay) it will be better to use transwell assay.

Ø  Why does the author use two different A375 human melanomas, MDA-MB-231 metastatic breast cancer cells?

Ø  The author needs to revise the (Fig. 3A, lower panels) (Fig. 3A, upper panels).

Ø  In line 457. What does it mean to “warm PBS”?

Ø  Mentioned the original dimension clearly in Figures 1 and 2.

Ø  In WB what type of “actin” does the author use as an antibody?

Ø  The author could possibly cite the article

DOI: 10.1016/j.phymed.2021.153500

Ø  Use EndNote or Mendeley software for reference sequences.

Ø  Check grammar and spelling throughout the manuscript. There are some mistakes.

Author Response

We would like to thank the reviewers for their time in reading our manuscript and for their comments. In general we found them constructive that improved our manuscript.

Reviewer 1

Ø  It would be better to include the quantification of WB in the figure in the entire manuscript. And also make the figure in consistent distance and size.

Response:

We want to point out that the vast majority of experiments presented as western blots are repeated is several figures. For example, the key findings of the manuscript including the effect of MLN4924 on the levels of Mdmx and RhoA are presented in almost every figure as independent experiments, showing the reproducibility of the results. Additionally, the results of other western blot analysis are confirmed by immunofluorescence analysis (Figure 5).

However, we agree in principle with the comment of the reviewer and we now quantified the intensity ratio in western blot analysis.

Ø  Section Introduction: Revise it and make it clear.

Response

We have improved the Introduction part even if we are not sure from the comment which part specifically we have to improve.

Ø  The author discussed “previous studies”. Why didn’t the author give the reference and need to revise and discuss the previous study and explain the previous study and this study’s result?

Response

We have now provided the appropriate references for all “previous studies”. We feel this should help the reader to see what has been previously done and what is new in the presented study.

Ø  Figure 4 and figure 6 WB results need to be repeated. The author needs to crop or fix the WB result.

Response

We apologise, but unfortunately, we do not understand the issue with figure 4 and 6 results. We believe the presented western blot analysis is clear. Some panels, for example Figure 4B, contain strong intensity bands of overexpressed RhoA. We think this is important for the readers to compare the relative expression between the overexpressed and endogenous RhoA.   

Mainor changes

Ø  Write keywords in alphabetical order.

Response

Corrected.

Ø  The authors want to put more latest related citations in the introduction part.

Response

Corrected. We have now added in the Introduction more recent-related citations.

Ø  In (4.7 Cell Invasion assay) it will be better to use transwell assay.

Response

Corrected. We now describe it as Transwell migration (Cell Invasion) assay

Ø  Why does the author use two different A375 human melanomas, MDA-MB-231 metastatic breast cancer cells?

Response

The major reason for this is that A375 and MDA-MB-231 cells are both metastatic but differ in the p53 status. A375 contain wild type p53 whereas MDA-MB-231 contain mutant p53. This was to assess if the p53 status plays a role in the Mdmx-RhoA signalling. This is now clearly stated in the text.

Ø  The author needs to revise the (Fig. 3A, lower panels) (Fig. 3A, upper panels). 

We apologise, but unfortunately, we do not understand what is the issue with Figure 3A, upper and lower panels.

Ø  In line 457. What does it mean to “warm PBS”? 

Response

We have now re-phrased this to PBS at 37oC.

Ø  Mentioned the original dimension clearly in Figures 1 and 2.

We apologise, but we do not understand what is the original dimension for figures 1 and 2.

Ø  In WB what type of “actin” does the author use as an antibody?

Response

We used a b-actin antibody that is now clearly stated.

Ø  The author could possibly cite the article 

DOI: 10.1016/j.phymed.2021.153500

Response

The article is now cited in the Introduction.

Ø  Use EndNote or Mendeley software for reference sequences.

Response

We have used Zotero software for the references.

Ø  Check grammar and spelling throughout the manuscript. There are some mistakes.

Response

We have now checked the manuscript for spelling mistakes.

Reviewer 2 Report

Comments and Suggestions for Authors

In this manuscript titled “An anti-invasive role for Mdmx through the RhoA GTPase under the control of the NEDD8 pathway”, by Malhab et al, they show the emerging tumor suppressor functions of Mdmx by controlling cell invasion through RhoA stabilization/activation.

This study is very interesting and written clearly.

Major comments:

Majority of this study is carried out using the A375 cell line. It would be interesting if the authors show different cell lines with varying expression levels of Mdmx.

It would be interesting if the authors show the status of p53 when silencing Ube2F and Ube2M.

Several figures lack statistical significance (p values are missing).

Author Response

We would like to thank the reviewers for their time in reading our manuscript and for their comments. In general we found them constructive that improved our manuscript.

Reviewer 2

In this manuscript titled “An anti-invasive role for Mdmx through the RhoA GTPase under the control of the NEDD8 pathway”, by Malhab et al, they show the emerging tumor suppressor functions of Mdmx by controlling cell invasion through RhoA stabilization/activation.

This study is very interesting and written clearly.

Major comments:

Majority of this study is carried out using the A375 cell line. It would be interesting if the authors show different cell lines with varying expression levels of Mdmx. 

We have used two different cell lines in this study, the malignant melanoma A375 and the breast adenocarcinoma MDA-MB-231 cells. The major reason is that they are both metastatic cells, which was an important aspect for our study, looking into the role of Mdmx in cell invasion. Additionally, these cell lines differ in the p53 status, A375 contains wild type p53 while MDA-MB-231 contains mutant p53. This allowed us to assess if the p53 status plays a role in the observed phenotypes.

We agree with the reviewer that looking into different cell lines with varying expression of Mdmx is an interesting and important point. However, we feel that such analysis may distract the readers from the key finding of our study on the presented role of Mdmx in cell invasion through the RhoA pathway. To note, we have observed the same effect of MLN4924 on Mdmx stability, in U2OS, HCT116 cells.

It would be interesting if the authors show the status of p53 when silencing Ube2F and Ube2M.

Response:

We have now re-analysed the extracts of the experiment presented in Figure 1B and present the effect of Ube2F/2M siRNA knockdown on p53 levels (Supplementary Figure S1). Knockdown of Ube2M increases p53 levels but Ube2F knockdown has no effect. This is consistent with the observation presented in Figure 1A, where inhibition of the NEDD8 pathway by MLN4924 increases p53 levels. The new data indicate that this effect is mediated mainly through the Ube2M E2 conjugating enzyme but not through Ube2F.

Several figures lack statistical significance (p values are missing).

Response:

We have now included the p-values for all presented quantitation experiments with the exception of the experiment in Figure 5A that represents only 2 independent experiments.